molecular biology

urea, membrane transporter, *Aspergillus nidulans*, codon bias

**Author for correspondence:**
Ana Ramón
e-mail: anaramon@fcien.edu.uy

[†]These authors contributed equally to this work.

### PUBLISHING

# A pair of non-optimal codons are necessary for the correct biosynthesis of the *Aspergillus nidulans* urea transporter, UreA

Manuel Sanguinetti[1], Andrés Iriarte[2,4,†],
Sotiris Amillis[3,†], Mónica Marín[1], Héctor Musto[4]
and Ana Ramón[1]

[1]Sección Bioquímica, Departamento de Biología Celular y Molecular, Facultad de Ciencias, Universidad de la República (UdelaR), Montevideo, Uruguay
[2]Laboratorio de Biología Computacional, Departamento de Desarrollo Biotecnológico, Instituto de Higiene, Facultad de Medicina, UdelaR, Montevideo, Uruguay
[3]Department of Biology, National and Kapodistrian University of Athens, Athens, Hellas, Greece
[4]Laboratorio de Organización y Evolución del Genoma, Unidad de Genómica Evolutiva, Departamento de Evolución, Facultad de Ciencias, UdelaR, Montevideo, Uruguay

MS, 0000-0002-4532-2383; AI, 0000-0002-5862-0152;
MM, 0000-0002-0162-9500; AR, 0000-0001-6810-6184

In both prokaryotic and eukaryotic genomes, synonymous codons are unevenly used. Such differential usage of optimal or non-optimal codons has been suggested to play a role in the control of translation initiation and elongation, as well as at the level of transcription and mRNA stability. In the case of membrane proteins, codon usage has been proposed to assist in the establishment of a pause necessary for the correct targeting of the nascent chains to the translocon. By using as a model UreA, the *Aspergillus nidulans* urea transporter, we revealed that a pair of non-optimal codons encoding amino acids situated at the boundary between the *N*-terminus and the first transmembrane segment are necessary for proper biogenesis of the protein at 37°C. These codons presumably regulate the translation rate in a previously undescribed fashion, possibly contributing to the correct interaction of *ureA*-translating ribosome-nascent chain complexes with the signal recognition particle and/or other factors, while the polypeptide has not yet emerged from the ribosomal tunnel. Our results suggest that the presence of the pair of non-optimal codons would not be functionally important in all cellular conditions. Whether this mechanism would affect other proteins remains to be determined.

# 1. Introduction

Most amino acids are encoded by more than one codon, each of which is usually used with unequal frequencies across the genomes of different organisms or even across a single gene, a phenomenon known as 'codon usage bias' [1,2]. Among genes, this phenomenon has been explained in terms of the balance between biases generated from mutation, natural selection and random genetic drift [3,4]. In several organisms, a close positive correlation has been established between gene expression levels and the frequencies of usage of certain codons, which usually match the most abundant transfer RNAs (tRNAs), this correlation being attributed to natural selection leading to fast and accurate translation [5–10]. This was shown to be the case for eight *Aspergillus* species for each of which, using available information on whole genome sequences and microarray data, a set of translational optimal codons could be defined [11].

Several lines of research suggest a role of codon usage in the control of translation initiation and elongation, as well as at the level of transcription and messenger RNA (mRNA) stability (for recent, excellent reviews see [12–15]). Translation rate control, in turn, has been related to the establishment of proper folding patterns, and hence functionality [15–25].

Owing to their complex structure and hydrophobic nature, polytopic membrane proteins pose an interesting challenge for the study of their folding mechanisms. In eukaryotes, these multispanning membrane proteins undergo a special biogenesis pathway through which they are co-translationally inserted into the endoplasmic reticulum (ER) membrane [26–29]. At early stages, during translation, the signal recognition particle (SRP) recognizes specialized signal sequences or hydrophobic motifs in peptides, which are destined for the membrane and installs a pause in the process. This pause is supposed to ensure an appropriate timing for the targeting of the translating ribosomes to the translocon, through which the different transmembrane segments are finally inserted into the ER membrane [30]. Besides the aforementioned influence of codon usage on translation elongation rates and hence folding and function, other roles specific to the special co-translational biosynthesis of membrane proteins have been disclosed. In *Escherichia coli*, *Saccharomyces cerevisiae* and *Aspergillus nidulans*, bioinformatic studies have revealed that membrane proteins are enriched in long, statistically significant, non-optimal codon clusters, which tend to occur about 45 and 70 codons after encoded hydrophobic stretches. These clusters have been suggested to impose pauses for correct membrane protein recognition and targeting (+45 cluster) and for correct insertion and folding (+70 cluster) [31,32]. Another study indicates that *N*-terminal positioned, long clusters of non-optimal codons, which are moreover conserved among evolutionary-related proteins, are found in membrane proteins and are proposed to be required for the proper interaction with other proteins needed for correct targeting and/or insertion [33]. The *in vivo* significance of codon usage in membrane protein biogenesis has been less explored. In the human multidrug resistance 1 gene *MDR1*, a single nucleotide polymorphism, resulting in a synonymous change, alters the conformation and function of the encoded polypeptide. The authors propose that this change presumably alters the timing of co-translational folding and insertion of the polypeptide into the membrane [34]. More recently, studies employing ribosome profiling in yeast revealed the presence of non-optimal codon clusters (REST elements, for mRNA-encoded slowdown of translation elements) placed 35–40 codons downstream of the region encoding the SRP-binding site. These REST elements were shown to enhance the recognition by SRP and hence to be necessary for the efficient translocation of membrane proteins [35].

In this study, we have developed a model that addresses the question of how codon usage could affect the folding, function and localization of membrane proteins, using the high-affinity urea/H$^+$ symporter of the ascomycete fungus *A. nidulans*, UreA, as a model [36,37]. We identified a pair of non-optimal, conserved codons coding for amino acids localized just at the boundary between the *N*-terminus of the protein and the predicted first helical transmembrane segment (TMS), whose optimization leads to a deficiency in UreA synthesis at the optimal growth temperature of 37°C, while at 25°C, the defect becomes much less severe. Our *in vivo* system supports the idea of an mRNA-encoded pause involved in the first steps of UreA synthesis and sorting to the membrane. The differences observed at the two assayed temperatures suggest that the relevance of this pausing event would depend on the cellular conditions playing on factors such as general translation rate, availability of folding chaperones and targeting machinery, etc.

# 2. Results

## 2.1. Identification of a pair of conserved, non-optimal codons in UreA and its orthologues

We reasoned that if UreA presents a codon-usage bias across its coding sequence, and if this bias has a role in protein expression and/or functionality, we could expect to find some synonymous codon usage

conservation between UreA and its orthologues in other Aspergilli. This conservation may not only consider the average frequency of usage of the gene but also the localization of optimal and non-optimal codons in specific regions of the gene in relation to the encoded protein structure.

Following this reasoning, the coding DNA sequences of the UreA orthologues in the eight *Aspergillus* species with known codon usage [11] were aligned, and the relative synonymous codon usage (RSCU) for each codon in highly expressed genes (HEGs) was determined. The RSCU is the ratio of the observed frequency of synonymous codons in a group of genes to the expected frequency, if all the codons coding for the same amino acid were used equally. It is a measure of the synonymous codon usage bias for each triplet, irrespectively of amino acid composition [3]. Thus, synonymous codons with RSCU values close to 1 are interpreted as not biased, that is, used as expected under null or marginal codon usage bias. RSCU values above 1 are interpreted as positive biased or used more than expected. In HEGs, these triplets are considered as translationally optimal codons, maintained by natural selection. It has been shown that these triplets are translated at higher speed and more accurately (reviewed by Sharp *et al.* [38]). Finally, RSCU values close to 0 are synonymous codons that are avoided in a certain group of genes.

A pair of 100% conserved, non-optimal and very rarely used triplets were identified in the eight Aspergilli. These correspond to codons 24 and 25 in the *A. nidulans ureA* sequence (figure 1). These two codons are $CAA_{(Gln)}$ and $GGG_{(Gly)}$ and have $RSCU_{(Total\ coding\ sequences,\ CDS)}$ values of 0.78 and 0.77, respectively, and $RSCU_{(HEGs)}$ values of 0.47 for CAA and 0.18 for GGG (see electronic supplementary material, table S1). After secondary structure predictions, performed with the TMHMM server (http://www.cbs.dtu.dk/services/TMHMM/) (see the electronic supplementary material, S1), we determined that in all of the orthologues, these two conserved non-optimal codons encode amino acidic residues lying at the boundary between the *N*-terminus of the protein and the predicted first TMS (figure 1).

## 2.2. The synonymous mutation of non-optimal codons 24 and 25 causes a defect in UreA's urea transport

To assess the functional relevance of conserved, non-optimal codons 24 and 25, we substituted them by their optimal synonymous counterparts, both individually and simultaneously. In the case of Gln(24), there is a single change option, so CAA was changed into CAG ($RSCU_{(Total\ CDS)} = 1.22$, $RSCU_{(HEGs)} = 1.56$). GGG coding for Gly(25) was substituted with GGT ($RSCU_{(Total\ CDS)} = 0.94$, $RSCU_{(HEGs)} = 1.83$). $CAG_{(Gln)}$ and $GGT_{(Gly)}$ triplets are significantly over-represented in HEGs and are considered as the unique putative optimal codons for the amino acid they encode [11] (see the electronic supplementary material, table S1).

Mutations were introduced by fusion polymerase chain reaction (Fusion-PCR) as described in Material and Methods. Genomic DNA from a strain carrying a wild-type (*wt*) UreA-green fluorescent protein (GFP) fusion was used as template, so that the subcellular localization of the resulting mutant UreA-GFP fusions can be easily assessed by epifluorescence microscopy. The constructs were transformed in a *ureAΔ* strain to obtain single copy transformants integrated into the *ureA locus* (*ureA2425* strains). Strains bearing *ureA::gfp* fusions behave exactly as wild-type strains in terms of growth on urea, urea transport and localization of the transporter to the membrane, indicating that these fusions are fully functional [36]. Mutant UreA functionality can be rapidly and simply evaluated by plate growth tests on urea or its toxic analogue 2-thiourea. Urea uptake capacity can be assessed by $^{14}$C-urea transport assays.

When codons 24 and 25 were individually mutated, the resulting mutant versions of *ureA* (strains *ureA24* and *ureA25*, respectively) did not show any differences in growth on urea or 2-thiourea at 37°C or 25°C, with respect to the *wt* (figure 2). Transport assay results were in agreement with this observation (figure 3*a*). Proteins bearing the single mutations are localized to the cell membrane in both cases, with no obvious changes in fluorescence levels (electronic supplementary material, figure S3).

However, when both non-optimal codons were mutated simultaneously (*ureA2425* mutant), growth at 37°C on 0.625–5 mM urea as a sole nitrogen source was intermediate between that of a *wt* and a *ureAΔ* strain. Accordingly, this mutant strain was partially resistant to the toxic urea analogue 2-thiourea, which is also transported into the cell through UreA [36,39]. It exhibited an intermediate phenotype between the complete resistance of the *ureA* knock-out strain and the sensitivity of the *wt*, which presents only a residual growth. At 25°C, however, the *ureA2425* strain showed phenotypes almost identical to that of the *wt*, both on urea and 2-thiourea (figure 2).

In accordance with growth tests, $^{14}$C-urea transport assays for *ureA2425* strains showed that the uptake rate of urea is approximately 50% lower than that of the *wt* strain at 37°C, while it reached

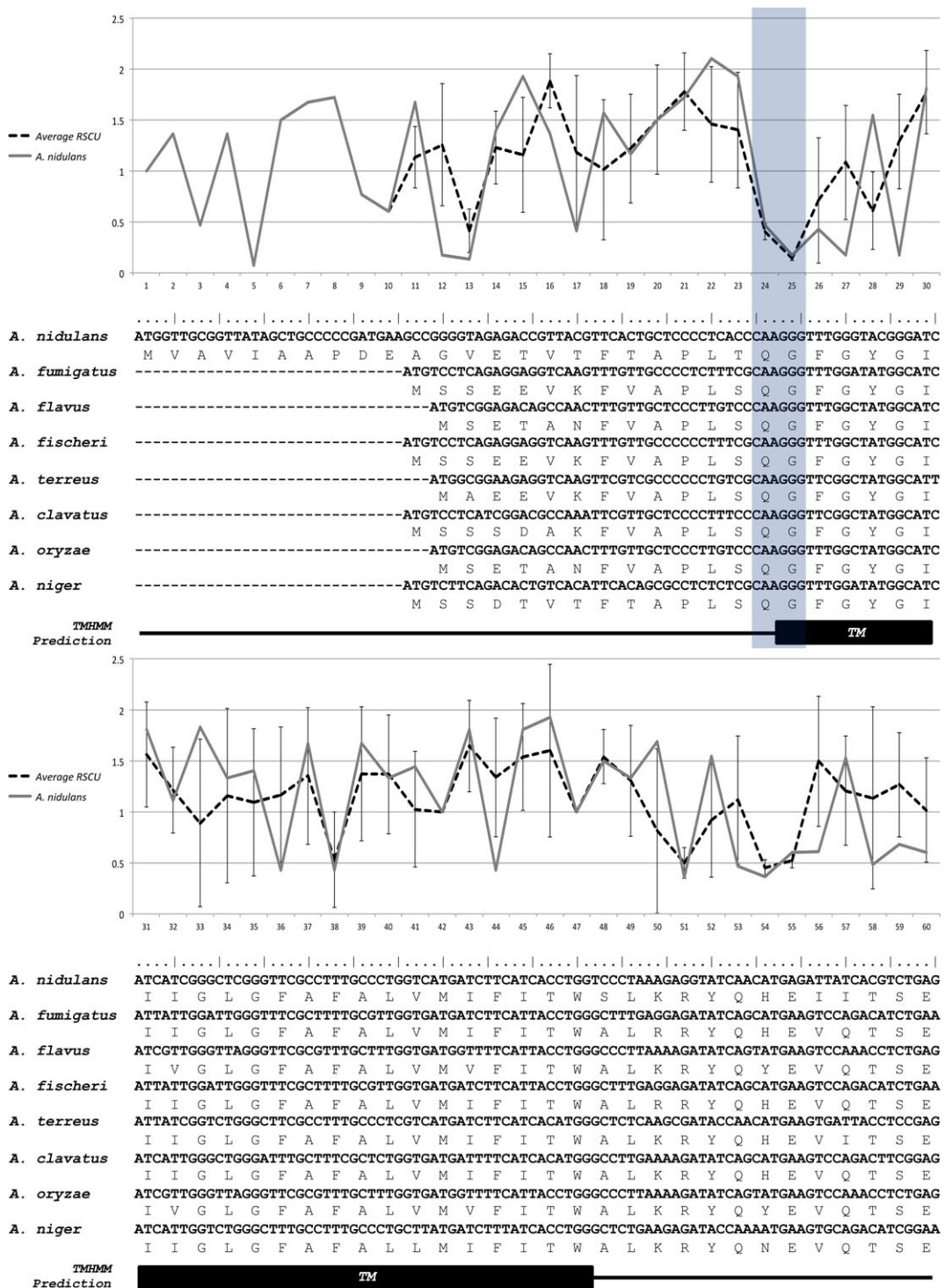

**Figure 1.** Partial alignment of UreA-coding sequences in eight *Aspergillus* species. Protein sequences were aligned and subsequently back-translated to the known nucleotide sequence. Relative synonymous codon usage in highly expressed genes (RSCU(HEGs)) is plotted for *A. nidulans* (continuous line), as well as the average of RSCU(HEGs) for the eight *Aspergillus* species (dotted line), with standard deviations. Note that negative values below 1 characterize synonymous triplets that are avoided in highly expressed genes and *vice versa*. Mutated codons 24 and 25 in *A. nidulans* are shadowed in grey. Prediction of the first transmembrane helix is shown as a black box below the alignment. Aligned sequences include *A. nidulans* AN0418, *A. fumigatus* AFUB_005210, *A. flavus* AFL2T_02167, *N. fischeri* NFIA_019890, *A. terreus* ATET_02629, *A. clavatus* ACLA_029800, *A. oryzae* AO_090003000854 and *A. niger* An01g03790 (http://www.aspergillusgenome.org).

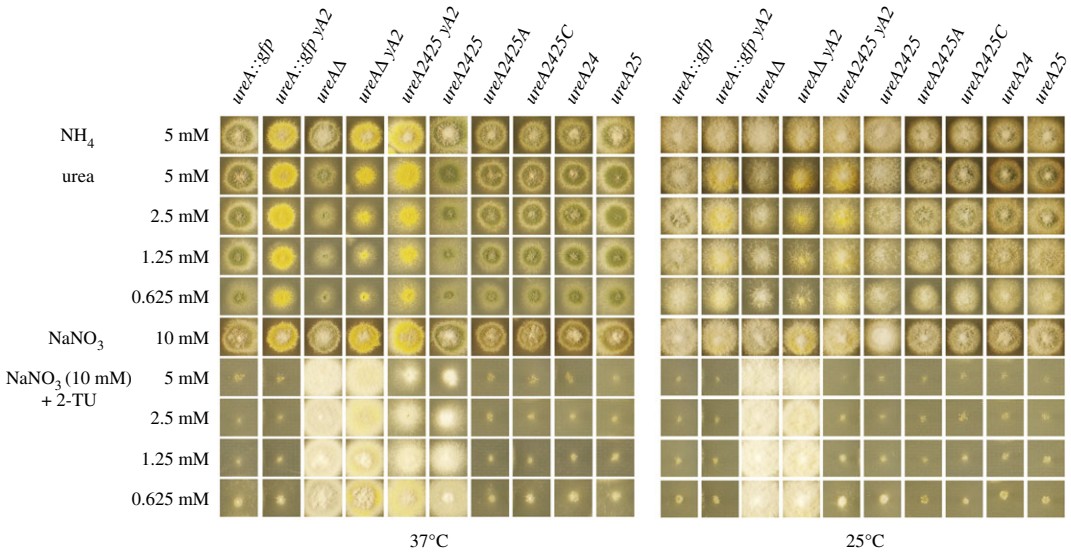

**Figure 2.** Characterization of strains bearing mutations in *ureA* codons 24 and 25. Growth phenotypes of mutant UreA strains at 25°C and 37°C on urea as nitrogen source or on 2-thiourea with sodium nitrate (NaNO₃) 10 mM as nitrogen source to test resistance to the analogue. Growths on 5 mM ammonium (NH₄) and 10 mM NaNO₃ are used as controls. A *wt ureA::gfp* and a *ureAΔ* strain are shown as positive and negative controls, respectively.

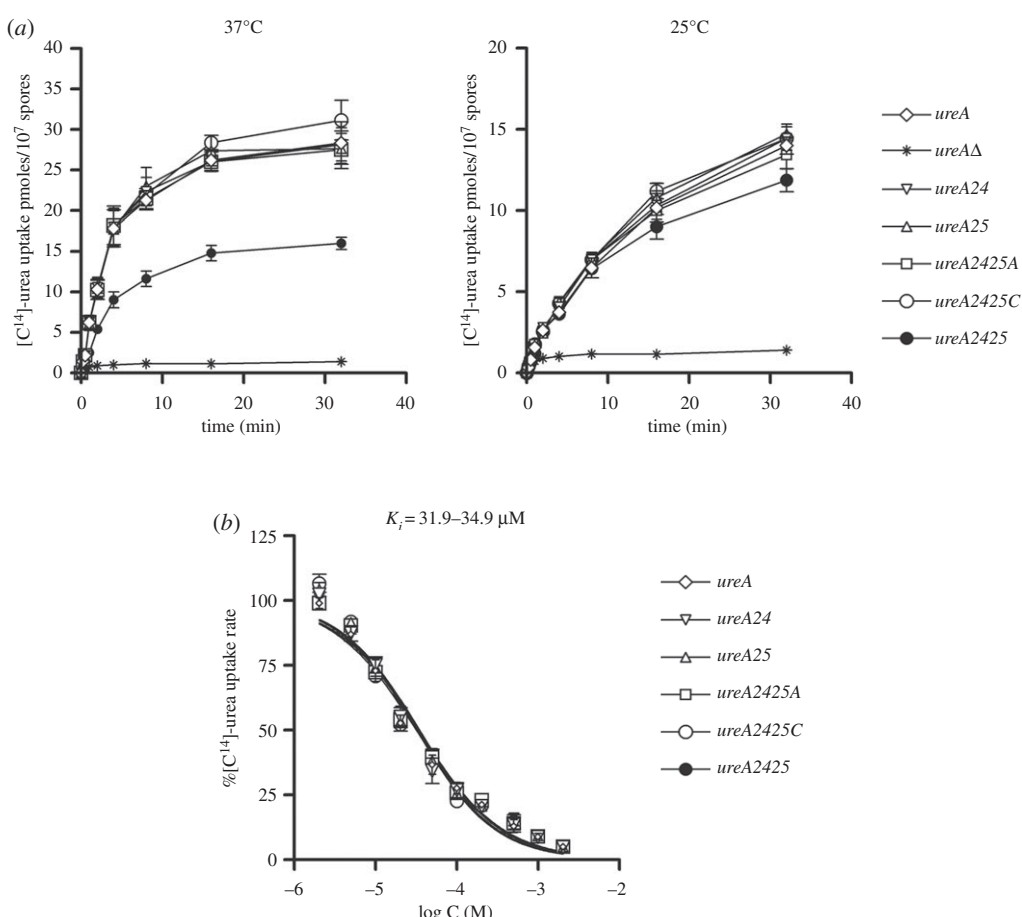

**Figure 3.** [¹⁴C]-urea uptake assays of strains bearing mutations in *ureA* codons 24 and 25. (*a*) Time course of [¹⁴C]-urea uptake, at 25°C and 37°C, in a wild-type strain (*ureA*), in a strain lacking the UreA transporter (*ureAΔ*) and in the different *ureA* codon mutants. (*b*) [¹⁴C]-urea uptake in a wild-type strain (*ureA*) and in the different *ureA* codon mutants in the presence of increasing concentrations of non-radiolabelled urea. The range of IC₅₀ values (equal to $K_{m/i}$) is depicted above (for details, see Materials and methods). Results are averages of three measurements for each concentration point.

more than 80% of the uptake rate of that of the *wt* at 25°C (figure 3*a*). By contrast, the measured substrate binding affinity of UreA2425 was similar to that of the *wt* (figure 3*b*).

Next, codon 25 was replaced with synonymous codons GGC and GGA (see above), in a *ureA24* context (optimal codon CAG in position 24), to obtain mutants *ureA2425C* and *ureA2425A*, respectively. According to the analysis of codon usage in *A. nidulans* [11], GGU would be the only optimal codon encoding Gly, so we would not expect any effects of changing GGG into GGA or GGC. Actually, no obvious differences could be detected in growth phenotypes (figure 2) or ${}^{14}$C-urea transport assays between the *wt* and the *ureA2425A* or *2425C* mutants (figure 3*a*).

We conclude that simultaneously changing non-optimal codons 24 and 25 into their predicted optimal synonyms causes impairment in the functional ability of UreA to transport urea.

## 2.3. the *ureA2425* mutation leads to a drastic decrease in UreA protein expression and localization to the plasma membrane

To follow the fate of GFP-tagged UreA2425 in the cell, we performed epifluorescence microscopy on mutant strain germlings (germinated fungal spores). In *wt* strains, grown at both 37°C and at 25°C on proline as a sole nitrogen source (derepressing conditions), UreA-GFP localized to the plasma membrane of germlings, to *septae* and, as a result of normal turnover, to intracellular globular compartments (vesicles and vacuoles) [36]. Accumulation of free GFP in the lumen of vacuoles, owing to its low-rate turnover in this compartment, can serve as an indicator of UreA-GFP degradation [40]. *ureA2425* strains grown at 37°C showed only a very faint florescence both in the cell membrane and in the globular intracellular compartments. If there was an augmented degradation of UreA2425-GFP, we would expect to observe GFP localized mainly in vacuoles. However, this was not the case. Detection of free GFP in western blots is also a standard, indirect measurement for vacuolar degradation of GFP-tagged membrane proteins [41,42]. Therefore, an increase in the free GFP signal should be observed if UreA2425-GFP was more prone to degradation. Notwithstanding, as shown in figure 4*b*, at 37°C, only faint signals were obtained in western blots for both UreA2425-GFP and free GFP. When *ureA2425::gfp* strains were grown at 25°C, UreA2425-GFP clearly localized in the cell membrane and in vacuoles, as in the *wt* (figure 4*a*). In agreement with these results, bands for both the fusion protein and free-GFP are clearly observed in western blots, even if UreA2425-GFP levels do not reach those of the *wt* (figure 4*b*).

As we cannot rule out the possibility that the mutant fusion protein is prone to proteasomal degradation, we carried out western blots on extracts of *wt* and *ureA2425* strains in the presence or absence of the proteasome inhibitors bortezomib and MG132, previously reported in *A. nidulans* [43,44]. No differences could be detected in UreA2425-GFP levels between the treated and the untreated samples (electronic supplementary material, figure S4a).

Our results suggest, then, that the low level of UreA2425-GFP in the membrane and the defective growth and transport of urea observed at 37°C could not be attributed to an augmented degradation of the mutant protein. Alternatively, the decrease in the amount of UreA2425-GFP levels could be explained as a defect in the synthesis of the protein. Membrane protein levels and hence transport are at least partially restored when the strain is grown at 25°C.

## 2.4. Changes in mRNA levels or structure at 37°C and 25°C do not explain the observed mutant phenotype

The impaired UreA2425 protein levels observed in western blot analyses raise the possibility that *ureA2425* mRNA levels are diminished in strains grown at 37°C. To evaluate this possibility, quantitative polymerase chain reaction (qPCR) was performed at both 37°C and 25°C, comparing *ureA* mRNA levels in mutant and *wt* strains grown on proline as a nitrogen source (figure 4*c*). Our results show that *ureA* mRNA levels are similarly diminished on the *ureA2425* strain with respect to the *wt* strain, at both 37°C and 25°C (66 ± 13.9% and 69 ± 5.6%, respectively). Notwithstanding, these mild differences would not explain the strong disparities observed at the level of protein synthesis at both temperatures.

As any change in sequence, a synonymous mutation could be at the basis of a change in RNA structure and/or structure stability. Stable secondary structures around translation initiation sequences and/or start codons in *ureA2425 mRNA* could influence the onset of translation. Thus, we performed a bioinformatic prediction on partial *ureA* and *ureA2425* mRNAs with mfold web server [45], on a

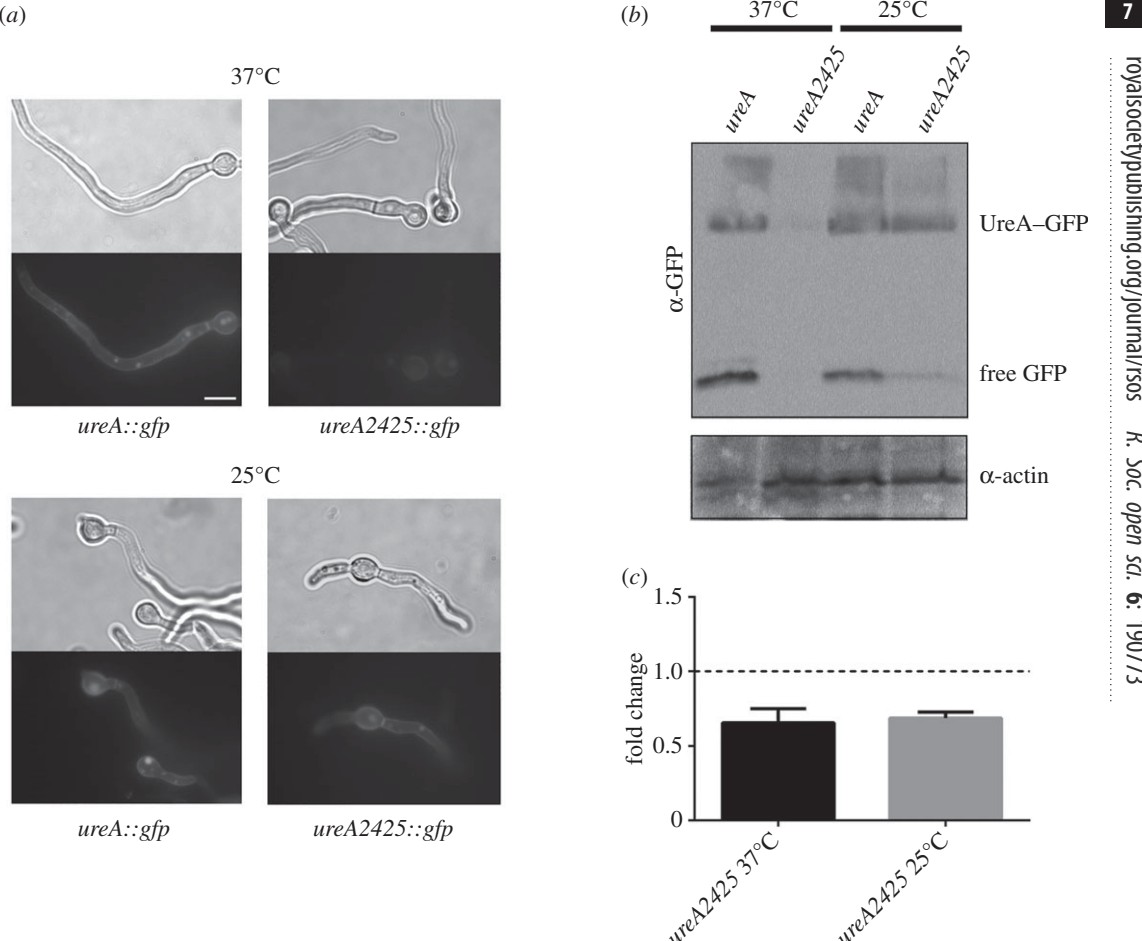

**Figure 4.** Expression of UreA2425-GFP. (*a*) Light microscope and epifluorescence images of mutant bearing the *ureA2425* mutation grown in derepressing conditions at 25 and 37°C. *wt* UreA–GFP localization is shown as a control. Scale bar, 10 μm. (*b*) Western blot analysis of total protein extracts of *ureA2425*–GFP mutants. Cultures were grown for 14–16 h at 25°C and 37°C in derepressing conditions (proline as a sole nitrogen source). Membranes were incubated with anti-GFP antibody (upper panel); the low mobility band corresponds to intact UreA–GFP, and the high mobility band corresponds to free GFP. Membranes were stripped and reincubated with anti-actin antibody (lower panel), as control of loading. Partial fields cropped from blots are shown; full-length blots are presented in the electronic supplementary material, figure S1. (*c*) Expression analysis of *ureA2425*, grown in derepressing conditions at 25 and 37°C, relative to expression of *ureA* in a *wt* strain (set as 1) is shown. Fold change was calculated using actin (*actA*) as a reference gene. The results are an average of three biological replicates, and error bars represent standard deviations.

sequence including 100 nucleotides upstream of the ATG start codon and 250 nucleotides downstream. No significant differences were detected in the predicted structures at 37°C and 25°C or in the minimum free energies (*ΔGs*) at both temperatures (electronic supplementary material, figure S2).

## 3. Discussion

In this study, we provide *in vivo* evidence of an important role of a pair of conserved, non-optimal codons in the biosynthesis and therefore activity of UreA. Simultaneous mutation of codons 24 and 25 into the corresponding optimal ones leads to a drastic diminution of UreA protein levels at the optimal growth temperature of 37°C. Such an effect could be owing to a decrease in mRNA levels or in protein synthesis, an increase in protein degradation or a combination of these possibilities.

Synonymous codon usage has been shown to participate in the regulation of mRNA stability [46–48]. On the other hand, as any mRNA sequence change, codon substitutions can affect the establishment of secondary, stable structural elements that may have an effect in the control of translation initiation and elongation [49–53]. Moreover, codon usage has been recently found to influence transcription levels in

*Neurospora crassa* by affecting chromatin structure [54]. We show here that in the *ureA2425* strain, mRNA levels suffer a decrease of approximately 30%, at both 37°C and at 25°C. Thus, our results suggest that codon usage could have some influence at the level of *ureA* transcription and/or mRNA stability. Notwithstanding, these differences at mRNA levels would not explain the striking decrease observed at 37°C in the levels of UreA in the *ureA2425* mutant.

Codon usage has been implicated in the control of protein folding and activity [16–18,21]. As mentioned above, in the case of membrane proteins, MDR1 constitutes an excellent example, where a mutant version bearing a single synonymous change presents altered protein conformation and functionality [34]. Notwithstanding, this does not seem to be the case for mutant UreA2425. First, our results on radiolabelled urea uptake measurements show a decrease in uptake rate, which is not accompanied by changes in the binding affinity, suggesting that the synonymous mutations would not severely affect the structure of at least functionally important parts of the protein. Second, the structure prediction for the first 25 amino acids of UreA suggests that this portion of the protein would be mainly unstructured (https://npsa-prabi.ibcp.fr/cgi-bin/npsa_automat.pl?page=/NPSA/ npsa_seccons.html). Third, misfolding of membrane proteins can lead to their retention in the ER, which in *A. nidulans* appears as a characteristic accumulation of fluorescence in perinuclear rings, as observed in the case of misfolded or partially misfolded UapA or UreA mutants [37,55,56]. In the case of UreA2425, no such localization could be detected. Retention of misfolded proteins in the ER can activate the UPR response and the expression of associated genes, like the BipA chaperone [55,57]. However, we could not detect significant differences in *bipA* mRNA levels between *wt* and *ureA2425* strains, determined by qPCR (electronic supplementary material, figure S4b). Finally, no indication of increased proteolysis of the mutant UreA2425 protein triggered by misfolding could be observed. We were unable to detect GFP accumulation in vacuoles nor an augmentation of free GFP in western blots. In addition, proteasome inhibition assays do not suggest that UreA2425-GFP is a substrate of this degradation complex.

Given all the above, we propose that the synonymous change of codons 24 and 25 must have a major impact at some stage of the protein synthesis process at 37°C. Considering the predicted localization of codons 24 and 25, at the boundary between the *N*-terminal portion of the protein and the predicted first TMS of UreA, it is reasonable to think that these non-optimal codons may be essential for a very early stage in the particular biogenesis pathway of membrane proteins. Our predictions of mRNA structure do not suggest an alteration at the level of secondary structure, which could lead to an early block in translation. An appealing alternative hypothesis is that the loss of the pair of non-optimal codons could affect a translational pause important for a key step in the biosynthesis of UreA. It is worth noticing that even if a direct relationship between codon usage and translation elongation speed has not been described for *A. nidulans*, such a relationship was demonstrated in *N. crassa* [25].

mRNA-encoded pauses have been suggested to have a role in assisting in the recognition of membrane proteins by cofactors necessary for their correct folding and insertion. During the translation of photosynthetic reaction centre protein Dl, a number of ribosome pauses occur, which may facilitate the co-translational binding of chlorophyll and aid the insertion of Dl into thylakoid membranes [58,59]. Recognition by SRP, a key player in membrane protein biogenesis, may also be regulated by pauses encoded in mRNA. In *E. coli*, the analysis of genome-wide data on translation rates revealed a tendency to pause in Shine-Dalgarno-like elements placed between codons 16 and 36, when the nascent polypeptide has not yet emerged out of the ribosomal exit tunnel. The authors suggest that at this early point, a slowdown in translation elongation would be important for early recognition by SRP and that it may compensate for the lack of Alu domain in prokaryotic SRP [60]. In yeast, the REST elements identified by Pechmann *et al.* [35] may assist in SRP recognition of the nascent polypeptide, by establishing a translational pause 35–40 codons downstream of the SRP-binding site. We wonder, then, if the pair of non-optimal codons 24 and 25 could establish a translational pause, an mRNA-encoded signal that may somehow contribute to the stabilization in the recognition of *ureA*-translating ribosome-nascent chain complexes (RNCs) by SRP or other factors required for early stages in the biosynthesis process of membrane proteins. It is important to note that such a putative translational pause would occur before the emergence of the polypeptide from the ribosomal tunnel [61]. In both prokaryotes and eukaryotes, it has been shown that SRP can stably bind to translating ribosomes harbouring short-chain peptides inside the exit tunnel, but it is not until the emergence of a signal sequence that further stabilization of the RNC-SRP interaction occurs [62–64]. Thus, what we observe in the case of *ureA* would be a novel mechanism, not described before in eukaryotes. As observed in *E. coli*, the putative arrest in translation would take place while the nascent polypeptide has not yet emerged from the ribosomal tunnel, but in the case of *ureA*, the

pause would be determined by a pair of non-optimal codons. This is quite different from Pechmann's results in *S. cerevisiae* [35], where the observed translational pause occurs 35–40 codons downstream of the SRP binding site, which would have then emerged from the ribosome.

It remains to be determined whether the presence of non-optimal codons in regions encoding the N-terminus of other *A. nidulans* membrane proteins would be necessary for their correct synthesis. In this sense, we performed a bioinformatic search of conserved, non-optimal codons in other *A. nidulans* membrane proteins, taking into account their position relative to the protein structure. We could only identify two other transporters where putative pauses would be found in a similar position as that in *ureA*. *hxtB*, encoding a low affinity glucose transporter [65], presents a non-optimal triplet encoding Asp at position 11. The first TMS is predicted to start at amino acidic residue number 11. *nrtA*, a high-affinity nitrate transporter [66], bears a conserved non-optimal codon in position 32, encoding a Tyr, with the first TMS starting at residue number 34, after secondary structure predictions.

We have shown that a single non-optimal codon of the 24/25 pair would be enough to grant a normal biosynthesis of UreA. We wonder, then, why both codons are conserved in the eight Aspergilli considered in this work. It has been found that codon pair usage is not random and has been shaped by translational selection, so that certain codon pairs are favoured or avoided in the three domains of life [67–71]. A possible explanation could be that the CAAGGG or the pairs formed by those codons with their neighbours would be among the favoured pairs, while those resulting from alternative triplets would be among the ones avoided. In fact, codon pairs containing the pattern nnGGnn are avoided [69]. Hence, because Gly25 could be encoded by GGA/C/G/T triplets, $CAG_{Gln}$ in position 24 would be unfavoured, while $CAA_{Gln}$ would be selected in this position.

Finally, the fact that at 25°C the *ureA2425* mutant displays an almost *wt* phenotype suggests that the role of non-optimal codons 24 and 25 would be dependent on cellular conditions. We can speculate that at the suboptimal growth temperature of 25°C, the translation would take place at a general lower rate, and thus, the putative pause in translation imposed by codons 24 and 25 would be less crucial than at 37°C. In this optimal growth temperature, general protein biogenesis rates would be higher, and the cell should need to adapt synthesis kinetics, in order not to saturate the translation and targeting machineries. Elongation pauses could then contribute to such a control mechanism.

To sum up, in this work, we report on *in vivo* evidence for the role of codon bias in membrane protein biogenesis. The pair of non-optimal codons 24 and 25 would regulate translation rate in a previously undescribed fashion, possibly contributing to the correct interaction of RNCs translating *ureA* with SRP and/or other factors necessary for these early events, while the polypeptide has not yet emerged from the ribosomal tunnel. Our results suggest that the presence of these non-optimal codons would not be functionally important in all cellular conditions. Whether this mechanism would affect other proteins is still to be determined.

# 4. Material and methods

## 4.1. Sequence analyses

Sequences were obtained from the Aspergillus Genome Database (http://www.aspgd.org/) [72]. Protein sequences were aligned using CLUSTAL OMEGA v.1.2.4 [73] and back translated to the known nucleotide coding sequence using the tranalign tool from the Emboss package [74]. The prediction of transmembrane regions was done using the TMHMM v.2.0 program [75]. Codon usage, including the estimation of the RSCU of coding sequences in the *A. nidulans* genome, was calculated using CODONW 1.4.4, written by John Peden and available at codonw.sourceforge.net. The RSCU of HEGs were taken as a proxy for the estimation of the bias towards the usage of translational optimal triplets. Thus, values above 1 in RSCU characterize triplets mainly associated with high speed and accuracy of translation [76], while values below 1 correspond to synonymous codons that show the opposite features and tend to be avoided in HEGs. mRNA secondary structure prediction was done using the mfold web server (http://unafold.rna.albany.edu/?q=mfold/RNA-Folding-Form [45]).

## 4.2. Strains, media and transformation procedures

Standard complete and minimal media (MM) for *A. nidulans* were employed ([77,78]; http://www.fgsc.net). Supplements were added when necessary at standard concentrations (http://www.fgsc.net/Aspergillus/gene_list/supplement.html). *Aspergillus nidulans* strains used in this study are listed in

the electronic supplementary material, table S2. Gene symbols are defined in http://www.fgsc.net/ Aspergillus/gene_list/loci.html. Urea (0.6–5 mM), NaNO₃ (10 mM), ammonium L(+)-tartrate (5 mM) and proline (5 mM) were used as sole nitrogen sources. 2-Thiourea was used in concentrations of 0.6–5 mM. *Aspergillus nidulans* transformation was carried out as in the study by Szewczyk *et al.* [79].

For experiments with proteasome inhibitors [43,44,56,80,81], extracts were prepared from 200 ml cultures of liquid MM with proline as a sole nitrogen source, inoculated with approximately $10^7$ spores ml$^{-1}$ and incubated at 37°C with shaking in the following conditions: (i) bortezomib was added from the beginning to an overnight culture, to a final concentration of 5 μM; (ii) cultures were grown overnight at 37°C, and after the addition of bortezomib to a final concentration of 5 μM, incubated for three extra hours; and (iii) cultures were grown for 20 h, before adding sodium dodecyl sulfate (SDS) to a final concentration of 0.003% and incubation for three additional hours. An MG132 solution in dimethyl sulfoxide (DMSO) was then added at a concentration of 120 μM, and incubation was proceeded for two additional hours. A culture added with the same volume of DMSO was used as a control.

## 4.3. Generation of *ureA* mutants

Site-directed mutagenesis on the *ureA::gfp* gene was performed by the fusion PCR technique [79] using KAPA HiFi DNA polymerase (KAPA Biosystems) with primers Ure5-F, Ure3-R and complementary ones carrying the desired substitution (see the electronic supplementary material, table S3). DNA extracted from a *ureA::gfp::AFpyrG* strain (MVD 10A) was used as a template. A resulting 7 kb fusion product was amplified with nested primers Ure5-N and Ure3-N and purified with the GeneJET Gel Extraction Kit (Thermo Scientific). The resulting construct was transformed in a *ureAΔ::riboB pyrG89 pyroA4 riboB2 nkuAΔ::argB veA1* strains (MVD 13A and isogenic *yA2* MVD 14A). Protoplasts were plated on selective (lacking uridine and uracil) 1 M sucrose MM and incubated at 37°C. Transformants were purified on selective MM, and its ability to grow in different concentrations of urea and 2-thiorea was tested. Transformants showing a *riboB2* phenotype were preferentially selected for further analysis as this was indicative of integration of the construct at the *ureA* locus. Assessment of single copy or multicopy transformants was performed by a DIG DNA labelling and detection kit (Roche Applied Science). Sequencing of the resulting mutants was performed in Macrogen Inc. (Seoul, Korea)

## 4.4. *ureA* expression analysis

Total RNA was isolated from *A. nidulans*, as described by Lockington *et al.* [82] from germlings grown on liquid cultures (using 5 mM proline as a nitrogen source) inoculated with $1 \times 10^7$ spores ml$^{-1}$ or $3 \times 10^7$ spores ml$^{-1}$ and incubated at 37 and 25°C, respectively. For cDNA synthesis, 1 μg total RNA was treated with Turbo DNase (Thermo Fisher Scientific) and retro-transcribed using SuperScript II Reverse Transcriptase (Invitrogen) in the presence of random primers (Invitrogen). Quantitative reverse transcription PCR was employed to determine the relative expression levels of *ureA* using *actA* (γ-actin) as an endogenous control for the calibration of gene expression. Specific primers (ureA_qPCR3_F/R and actA_F/R, electronic supplementary material, table S3) were used to amplify the corresponding cDNA fragments. PCR reactions were performed using the SensiFast SYBR Hi-ROX Kit (Bioline) in a StepOne Real-Time PCR System (Thermo Fisher Scientific). Three biological and two technical replicates for each sample were performed. Relative gene expression data were analysed using the $2^{\Delta\Delta Ct}$ method [83].

## 4.5. Protein extraction and western blot

MM liquid cultures with proline (5 mM) as a nitrogen source were incubated overnight at 37°C or 25°C. Micelia was then harvested and ground in liquid nitrogen. Ground mycelium (200 mg) was suspended in precipitation buffer (50 mM Tris–HCl pH 8.0, 50 mM NaCl and 12.5% TCA) and centrifuged for 10 min at 16 000*g*. Pellets were recovered and suspended in extraction buffer (100 Mm Tris–HCl pH 8.0, 50 mM NaCl. 1% SDS, 1 mM EDTA and 1:250 dilution of Sigma's fungal protease inhibitor) and centrifuged for 15 min at 16 000 g. The supernatant was recovered, and protein concentration was determined by the Bradford assay. Total proteins (30 μg) were separated by SDS-polyacrylamide gel electrophoresis (10% (w/v) polyacrilamide gel) and electroblotted (Mini Trans-Blot Electrophoretic Transfer Cell, Bio-Rad) onto a nitrocellulose membrane (0.45 μM, Thermo Scientific). Transferred proteins on the membrane were incubated in stripping buffer (10 mM Tris–HCl pH 6.8, 2% SDS and 100 β-mercaptoethanol) for

15 min at 55°C (as described by Kaur & Bachhawat [84]). The membrane was then treated with 5% non-fat dry milk, and immunodetection was done with a primary mouse anti-GFP monoclonal antibody (Roche Applied Science) or an anti-actin monoclonal antibody (C4, MP Biomedicals) and a secondary sheep anti-mouse IgG HRP-linked antibody (GE Healthcare). Blots were developed using the Novex ECL HRP Chemiluminescent Substrate Reagent kit (Invitrogen) and the GENESYS software from the GBox Chemi XT4 System (Syngene) or SuperRX Fuji medical X-ray films (FujiFILM).

## 4.6. Epifluorescence microscopy

Samples for fluorescence microscopy were prepared as described previously [85]. In brief, samples were incubated directly on sterile coverslips protected from light in liquid MM with proline (5 mM) as a nitrogen source and appropriate supplements at 25°C for 14–16 h or 37°C for 7–8 h. Samples were observed and photographed in an Olympus inverted microscope CKX31 belonging to the Cellular Biology Platform, Institut Pasteur de Montevideo, with a U-MNIBA3 filter. The microscope is equipped with a Hamamatsu Orca Er camera and uses MICRO-MANAGER software (https://micro-manager.org/, [86]) for image processing. The resulting images were processed by Adobe PHOTOSHOP software.

## 4.7. Radiolabelled urea uptake measurements

[$^{14}$C]-urea uptake was performed, as previously described, in germinating conidiospores of *A. nidulans* grown for 3–4 h in liquid MM (37°C, 130 r.p.m., pH 6.8), supplemented with 1% (w/v) glucose and 10 mM proline as carbon and nitrogen sources, respectively, concentrated at approximately $10^7$ conidiospores/assay [36,87]. In time course experiments, uptake was measured in the presence of 50 μM [$^{14}$C]-urea. Reactions were terminated by the addition of excess non-labelled substrate (1000-fold). Kinetic values ($Km/i$) were obtained by performing and analysing uptakes (PRISM 3.02: Graph PadSoftware, Inc.), as described in detail by Abreu *et al.* [36] and Sanguinetti *et al.* [37]. Assays were carried out in at least three independent experiments, with three replicates for each time point. Background uptake was corrected by subtracting values measured in the deleted mutant (*ureAΔ*). Data were analysed and plotted using the PRISM software (PRISM 3.02: Graph Pad Software). The standard deviation was less than 20%. [$^{14}$C]-urea (55.0 mCi mmol$^{-1}$) was purchased from Moravek Biochemicals (Brea, CA, USA).

Data accessibility. Data is available from the Dryad Digital Repository at https://doi.org/10.5061/dryad.56kv356 [88]. Data is also available from electronic supplementary material, figures and tables. *Aspergillus nidulans* strains used in this study are available upon request.
Authors' contribution. A.R., M.M., H.M. and A.I. designed the experiments. A.I. developed bioinformatics resources; A.I., A.R. and M.S. performed bioinformatics analyses. S.A. performed transport assays. M.S. and A.R. constructed mutant strains and carried out microscopy studies and western blots. M.S. performed qPCR and mRNA secondary structure predictions. M.S., A.I., S.A., M.M., H.M. and A.R. carried out results analysis and discussion. A.R. supervised the project and wrote the manuscript with contributions from the other authors.
Competing interests. The authors declare no financial and/or non-financial competing interests.
Funding. This work was supported by Agencia Nacional de Investigación e Innovación, Uruguay grant no. FCE_2009_12776. M.S. was the recipient of fellowships from Agencia Nacional de Investigación e Innovación (ANII, Uruguay) and a PhD fellowship from Comisión Académica de Posgrado, UdelaR, Uruguay. A.R., M.S., H.M., M.M. and A.I. are members of the National Research System (SNI, Sistema Nacional de Investigadores).
Acknowledgements. S.A. thanks G. Diallinas for providing the [$^{14}$C] experimental facilities. We thank Ricardo Ehrlich for critical reading of this work. Bortezomib was a gift from D. Stravopodis, to whom we are grateful.

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
