## [Reviewer comments · Royal Society Open Science]

Review History

RSOS-190773.R0 (Original submission)

Review form: Reviewer 1 (Tobias von der Haar)

Is the manuscript scientifically sound in its present form?

Yes

Are the interpretations and conclusions justified by the results?

Yes

Is the language acceptable?

Yes

Is it clear how to access all supporting data?

Not Applicable

Do you have any ethical concerns with this paper?

No

Have you any concerns about statistical analyses in this paper?

No

Recommendation?

Accept with minor revision (please list in comments)

Comments to the Author(s)

The manuscript by Sanguinetti et al describes a study into codon usage in an *Aspergillus* membrane protein. The authors identify a particular pair of slowly decoded codons preceding the first transmembrane protein in *UreA* as conserved non-optimal codons, and then show by mutating these codons to other, synonymous codons that having at least one non-optimal codon in this pair is essential for full expression and activity of the transporter.

Overall I find this an interesting paper that enriches the largely theoretical literature on codon usage and protein functionality with some experimental work. The authors have characterised the codon mutants fairly exhaustively, and have been careful in their conclusions. However, I do think that the discussion around the connection between the two codons and folding is a little superficial, as detailed below, and some aspects of the presentation could be enhanced a little.

1) My biggest issue is with the conclusion of the authors that the codons do not actually interfere with folding. Their data certainly show no proof for difficulties with folding and in this their discussion is correct, but I do not think they can completely exclude defective folding based on their data either. To my knowledge the assay based on detection of free vacuolar GFP works for cytoplasmically expressed GFP-fused proteins which are degraded via autophagy-like mechanisms, but it does not work eg for proteasomal targets. I think here the discussion should be improved.

A more direct experiment could be to directly measure stability of the *UreA*-GFP fusion following addition of cycloheximide to the culture – is this feasible in *Aspergillus*?

2) The fact that the slow codons are conserved in the eight *Aspergillus* species is fairly central to the story of the manuscript, and I think details should be shown here. Rather than showing RSCU for *A. nidulans* in figure 1, it would be better to show the average RSCU for the eight species.

3) I do not quite follow the rationale for selecting the codon pair at location 24/25, since there actually seems to be a sequence of four poorly adapted codons in a row – can the authors clarify this?

4) Line 125, “revised” by sharp et al. should be “reviewed” by Sharp et al.

5) Line 197/198, should “non-repressing” better be “permissive”?

6) Line 228/229, “mRNA levels are similarly diminished in the mutant strain” it should be clarified which of the mutant strains is referred to.

Review form: Reviewer 2

Is the manuscript scientifically sound in its present form?

No

Are the interpretations and conclusions justified by the results?

No

Is the language acceptable?

No

Is it clear how to access all supporting data?

Not Applicable

Do you have any ethical concerns with this paper?

No

Have you any concerns about statistical analyses in this paper?

No

Recommendation?

Major revision is needed (please make suggestions in comments)

Comments to the Author(s)

In this manuscript, Manuel Sanguinetti and co-authors attempted to demonstrate that a pair of nonoptimal codons, encoding Gln and Gly, at the junction of the N-terminal fragment and the first transmembrane helical segment of *Aspergillus nidulans* urea transporter, UreA, is important for the biosynthesis of UreA and, presumably, its proper folding. The authors hypothesized that this pair of nonoptimal codons is essential for proper co-translational targeting of transmembrane segment(s) into ER membrane.

The authors tested their hypothesis by i) mutating these codons (individually or simultaneously) into synonymous optimal ones, ii) measuring UreA mRNA and protein levels, as well as urea uptake. The authors found that mutation of just one of the two codons doesn't affect UreA biogenesis, while simultaneous mutation of the two codons affects UreA biogenesis. The observed effects were more pronounced at 37 deg C, but not at 24 deg C.

The authors argued that the pair of non-optimal codons "would presumably regulate translation rate in a very early stage in the process of biosynthesis, maybe contributing to the correct interaction of RNCs translating ureA with SRP and/or other factors necessary for these early events."

This study is interesting, although not novel and far too preliminary to be published in its present form.

Major:

- (1) Apart from the "+70 pause hypothesis" originally suggested by François Képès 1996, it has been shown by Mullet and co-authors that ribosomes pause at specific sites during synthesis of membrane-bound chloroplast reaction center protein D1 and this pausing may facilitate co-translational folding of D1 and aid the integration of D1 into thylakoid membranes (see Kim et al, J Biol Chem. 1991; Kim et al, J Biol Chem. 1994, etc). The authors must cite these original observations.
- (2) The authors apparently presumed that synonymous substitutions under study would not lead to miscoding. While true in many cases, such possibility couldn't be excluded (see Drummond and Wilke, Cell 2008; Buhr et al. Mol Cell, 2016, etc). The authors thus have to exclude miscoding by microsequencing of the silently mutated products.
- (3) Was the interaction with SRP indeed affected, as the authors hypothesized?
- (4) The origin of the drastic reduction of the levels of the silently mutant UreA variant remains unclear and has to be determined.
- (5) The authors also have to appropriately cite recent reports from Dr. Elizabeth Grayhack's lab (see e.g. Ghoneim et al. Conservation of location of several specific inhibitory codon pairs in the *Saccharomyces sensu stricto* yeasts reveals translational selection. Nucleic Acids Res. 2019; Gamble et al. Adjacent Codons Act in Concert to Modulate Translation Efficiency in Yeast. Cell. 2016).

(5) Some other recent reviews on the function of the rare codon clusters should be mentioned as well (the Chaney JL, Clark PL. Roles for Synonymous Codon Usage in Protein Biogenesis. *Annu Rev Biophys.* 2015; Komar AA. The Yin and Yang of codon usage. *Hum Mol Genet.* 2016).

Minor:

1. The authors should attempt to write the paper in a more concise and clear way and carefully proofread it.
2. While describing the location of the rare codon cluster under study (within the UreA mRNA), the authors should use the term "boundary" (as they did on page 13; line 279), or border, or junction, rather than "limit".
3. Page 7, line 125: "(revised by Sharp et al. [37])." I am assuming the authors meant to write "reviewed by".

Decision letter (RSOS-190773.R0)

17-Jul-2019

Dear Dr Ramon,

The editors assigned to your paper ("A pair of nonoptimal codons are necessary for the correct biosynthesis of the *Aspergillus nidulans* urea transporter, UreA") have now received comments from reviewers. We would like you to revise your paper in accordance with the referee and Associate Editor suggestions which can be found below (not including confidential reports to the Editor). Please note this decision does not guarantee eventual acceptance.

Please submit a copy of your revised paper before 09-Aug-2019. Please note that the revision deadline will expire at 00.00am on this date. If we do not hear from you within this time then it will be assumed that the paper has been withdrawn. In exceptional circumstances, extensions may be possible if agreed with the Editorial Office in advance. We do not allow multiple rounds of revision so we urge you to make every effort to fully address all of the comments at this stage. If deemed necessary by the Editors, your manuscript will be sent back to one or more of the original reviewers for assessment. If the original reviewers are not available, we may invite new reviewers.

- Data accessibility

<http://datadryad.org/submit?journalID=RSOS&manu=RSOS-190773>

- Competing interests

- Authors' contributions

- Acknowledgements

- Funding statement

Kind regards,

on behalf of Dr Stephen Long (Associate Editor) and Catrin Pritchard (Subject Editor)
openscience@royalsociety.org

Comments to Author:

Reviewers' Comments to Author:

Reviewer: 1

Comments to the Author(s)

The manuscript by Sanguinetti et al describes a study into codon usage in an *Aspergillus* membrane protein. The authors identify a particular pair of slowly decoded codons preceding the first transmembrane protein in *UreA* as conserved non-optimal codons, and then show by mutating these codons to other, synonymous codons that having at least one non-optimal codon in this pair is essential for full expression and activity of the transporter.

Overall I find this an interesting paper that enriches the largely theoretical literature on codon usage and protein functionality with some experimental work. The authors have characterised the codon mutants fairly exhaustively, and have been careful in their conclusions. However, I do think that the discussion around the connection between the two codons and folding is a little superficial, as detailed below, and some aspects of the presentation could be enhanced a little.

1) My biggest issue is with the conclusion of the authors that the codons do not actually interfere with folding. Their data certainly show no proof for difficulties with folding and in this their discussion is correct, but I do not think they can completely exclude defective folding based on their data either. To my knowledge the assay based on detection of free vacuolar GFP works for cytoplasmically expressed GFP-fused proteins which are degraded via autophagy-like mechanisms, but it does not work eg for proteasomal targets. I think here the discussion should be improved.

A more direct experiment could be to directly measure stability of the *UreA*-GFP fusion following addition of cycloheximide to the culture – is this feasible in *Aspergillus*?

2) The fact that the slow codons are conserved in the eight *Aspergillus* species is fairly central to the story of the manuscript, and I think details should be shown here. Rather than showing RSCU for *A. nidulans* in figure 1, it would be better to show the average RSCU for the eight species.

3) I do not quite follow the rationale for selecting the codon pair at location 24/25, since there actually seems to be a sequence of four poorly adapted codons in a row – can the authors clarify this?

4) Line 125, “revised” by sharp et al. should be “reviewed” by Sharp et al.

5) Line 197/198, should “non-repressing” better be “permissive”?

6) Line 228/229, “mRNA levels are similarly diminished in the mutant strain” it should be clarified which of the mutant strains is referred to.

Reviewer: 2

Comments to the Author(s)

In this manuscript, Manuel Sanguinetti and co-authors attempted to demonstrate that a pair of nonoptimal codons, encoding Gln and Gly, at the junction of the N-terminal fragment and the first transmembrane helical segment of *Aspergillus nidulans* urea transporter, UreA, is important for the biosynthesis of UreA and, presumably, its proper folding. The authors hypothesized that this pair of nonoptimal codons is essential for proper co-translational targeting of transmembrane segment(s) into ER membrane.

The authors tested their hypothesis by i) mutating these codons (individually or simultaneously) into synonymous optimal ones, ii) measuring UreA mRNA and protein levels, as well as urea uptake. The authors found that mutation of just one of the two codons doesn't affect UreA biogenesis, while simultaneous mutation of the two codons affects UreA biogenesis. The observed effects were more pronounced at 37 deg C, but not at 24 deg C.

The authors argued that the pair of non-optimal codons “would presumably regulate translation rate in a very early stage in the process of biosynthesis, maybe contributing to the correct interaction of RNCs translating ureA with SRP and/or other factors necessary for these early events.”

This study is interesting, although not novel and far too preliminary to be published in its present form.

Major:

(1) Apart from the “+70 pause hypothesis” originally suggested by François Képès 1996, it has been shown by Mullet and co-authors that ribosomes pause at specific sites during synthesis of membrane-bound chloroplast reaction center protein D1 and this pausing may facilitate co-translational folding of D1 and aid the integration of D1 into thylakoid membranes (see Kim et al, J Biol Chem. 1991; Kim et al, J Biol Chem. 1994, etc). The authors must cite these original observations.

(2) The authors apparently presumed that synonymous substitutions under study would not lead to miscoding. While true in many cases, such possibility couldn't be excluded (see Drummond and Wilke, Cell 2008; Buhr et al. Mol Cell, 2016, etc). The authors thus have to exclude miscoding by microsequencing of the silently mutated products.

(3) Was the interaction with SRP indeed affected, as the authors hypothesized?

(4) The origin of the drastic reduction of the levels of the silently mutant UreA variant remains unclear and has to be determined.

(5) The authors also have to appropriately cite recent reports from Dr. Elizabeth Grayhack's lab (see e.g. Ghoneim et al. Conservation of location of several specific inhibitory codon pairs in the *Saccharomyces sensu stricto* yeasts reveals translational selection. Nucleic Acids Res. 2019; Gamble et al. Adjacent Codons Act in Concert to Modulate Translation Efficiency in Yeast. Cell. 2016).

(5) Some other recent reviews on the function of the rare codon clusters should be mentioned as well (the Chaney JL, Clark PL. Roles for Synonymous Codon Usage in Protein Biogenesis. Annu Rev Biophys. 2015; Komar AA. The Yin and Yang of codon usage. Hum Mol Genet. 2016).

Minor:

1. The authors should attempt to write the paper in a more concise and clear way and carefully proofread it.
2. While describing the location of the rare codon cluster under study (within the UreA mRNA), the authors should use the term “boundary” (as they did on page 13; line 279), or border, or junction, rather than “limit”.
3. Page 7, line 125: “(revised by Sharp et al. [37]).” I am assuming the authors meant to write “reviewed by”.

Author's Response to Decision Letter for (RSOS-190773.R0)

See Appendix A.

RSOS-190773.R1 (Revision)

Review form: Reviewer 2

Is the manuscript scientifically sound in its present form?

Yes

Are the interpretations and conclusions justified by the results?

Yes

Is the language acceptable?

Yes

Do you have any ethical concerns with this paper?

No

Have you any concerns about statistical analyses in this paper?

No

Recommendation?

Accept as is

Comments to the Author(s)

The manuscript by Manuel Sanguinetti and co-authors has been revised. Additional information has been added, which was missing in the original version of the manuscript. In sum, I feel that the authors have responded to the majority of the previous concerns either through additional experiments or changes to the text.

Decision letter (RSOS-190773.R1)

07-Oct-2019

Dear Dr Ramon,

I am pleased to inform you that your manuscript entitled "A pair of nonoptimal codons are necessary for the correct biosynthesis of the *Aspergillus nidulans* urea transporter, UreA" is now accepted for publication in Royal Society Open Science.

on behalf of Dr Stephen Long (Associate Editor) and Catrin Pritchard (Subject Editor)
openscience@royalsociety.org

Reviewer comments to Author:
Reviewer: 2

Comments to the Author(s)

The manuscript by Manuel Sanguinetti and co-authors has been revised. Additional information has been added, which was missing in the original version of the manuscript. In sum, I feel that the authors have responded to the majority of the previous concerns either through additional experiments or changes to the text.

Follow Royal Society Publishing on Twitter: [@RSocPublishing](https://twitter.com/RSocPublishing)

Appendix A

Responses to the reviewer's comments are indicated in bold

We would like to thank the reviewer for constructive comments and suggestions. Answers to specific points follow below.

Reviewer 1

The manuscript by Sanguinetti et al describes a study into codon usage in an *Aspergillus* membrane protein. The authors identify a particular pair of slowly decoded codons preceding the first transmembrane protein in UreA as conserved non-optimal codons, and then show by mutating these codons to other, synonymous codons that having at least one non-optimal codon in this pair is essential for full expression and activity of the transporter.

Overall I find this an interesting paper that enriches the largely theoretical literature on codon usage and protein functionality with some experimental work. The authors have characterized the codon mutants fairly exhaustively, and have been careful in their conclusions. However, I do think that the discussion around the connection between the two codons and folding is a little superficial, as detailed below, and some aspects of the presentation could be enhanced a little.

1) My biggest issue is with the conclusion of the authors that the codons do not actually interfere with folding. Their data certainly show no proof for difficulties with folding and in this their discussion is correct, but I do not think they can completely exclude defective folding based on their data either. To my knowledge the assay based on detection of free vacuolar GFP works for cytoplasmically expressed GFP-fused proteins which are degraded via autophagy-like mechanisms, but it does not work eg for proteasomal targets. I think here the discussion should be improved. A more direct experiment could be to directly measure stability of the UreA-GFP fusion following addition of cycloheximide to the culture – is this feasible in *Aspergillus*?

Attending to the reviewer's concerns, we have performed a series of experiments which we hope will make our conclusions more convincing:

- a) **We have done a pulse chase with cycloheximide at 25 and 32°C (the very low level of UreA2425 protein observed at 37°C precludes to perform this type of analysis at this temperature) but the results do not provide further information on the stability of wild type and/or mutant UreA-GFP. As you can see in the figure R1 below, the stability of the protein is such, in both cases, that after 4 hours of incubation with cycloheximide no real decay of the protein is observed. This has been also shown to be the case for other *Aspergillus* membrane proteins⁽¹⁾, which would seem to be very stable, in general. It is important to note that after some time the proteins involved in the endocytic route must also be degraded. In the UreA characterization paper we published in 2010⁽²⁾, we showed in a microscopy assay, that after a 2-hour treatment in the presence of cycloheximide, UreA-GFP could not be internalized in conditions promoting its degradation, remaining at the membrane. This result suggests the need of protein synthesis for degradation of UreA-GFP.**
- b) **As explained in the manuscript, misfolding could result in the accumulation and retention of UreA-GFP in the endoplasmic reticulum, with the characteristic localization in perinuclear rings, which we do not observe⁽³⁾. In order to complement this result we determined the levels of an unfolded protein response (UPR)-associated gene, *bipA*, by qPCR (see figure R2, added as Supplementary material figure S4b). The absence of significant differences in *bipA* mRNA levels between the *wt* and the *ureA2425* strain would exclude the possibility of UPR activation in response to the accumulation of misfolded proteins. These results have been discussed in the revised version of the manuscript.**
- c) **In order to exclude proteasomal degradation we performed Western blots on *wt* and *ureA2425* strains at 37 °C, in the presence or absence of two different proteasome inhibitors: bortezomib and MG132. The results, shown in Supplementary figure S4a, do not show any augmentation of UreA2425-GFP levels, as would be expected if the protein, as a consequence of misfolding, was subjected to proteasomal degradation. These results were included in the manuscript, both in Results and Discussion.**

Finally, we signal that even if the uptake rate of the mutant protein is diminished, no significant alterations in the substrate binding affinity was observed, which would indicate that at least functionally important parts of the protein are not structurally altered.

Figure R1. (A) Cycloheximide (CHX) pulse chase at 25°C. *ureA* wild type and *ureA2425* strains were cultured on proline (5 mM) as sole nitrogen source for 20 hours at 25°C. A pulse chase with CHX at a concentration of 100 µg/mL was done at time 0. Mycelia were recovered for protein extraction in 30 minutes intervals for 2 hours. Western blots were performed with anti-GFP antibody and, as a loading control, an actin antibody. In the left panel, the two lanes to the right of the molecular weight marker (MWM), signaled CHX(+) and CHX (-) are controls of the effect of cycloheximide at the level of protein synthesis in *Aspergillus nidulans*. After 20 hours of culture of a wild type strain on ammonium (10 mM), where *ureA* transcription is repressed, the mycelia were transferred to non-repressing conditions (media lacking nitrogen source) for 2 hours in the presence (+) or absence (-) of CHX. The inhibition of translation can be seen in CHX (+) conditions. **(B) Cycloheximide pulse chase at 32°C.** As protein synthesis of UreA2425 is very low at 37°C, this assay was done at 32°C (intermediate temperature between 25 and 37°C). Cultures and Western blots were performed as described in Fig. 1, except that cultures were followed after CHX treatment for up to 4,5 hours.

*the lane corresponding to t=60 min in the *wt* strain must be ignored, since there was an error in sample preparation

Figure R2. Relative expression of *bipA* (AN2062) in *ureA2425* relative to the expression in a *wt* strain (set as 1). Strains were grown in derepressing conditions (proline) at 37°C. Fold change was calculated using actin (*actA*) as reference gene.

2) The fact that the slow codons are conserved in the eight *Aspergillus* species is fairly central to the story of the manuscript, and I think details should be shown here. Rather than showing RSCU for *A. nidulans* in figure 1, it would be better to show the average RSCU for the eight species.

Figure 1 was modified according to this suggestion

3) I do not quite follow the rationale for selecting the codon pair at location 24/25, since there actually seems to be a sequence of four poorly adapted codons in a row – can the authors clarify this?

Selected codons were chosen on the basis of the following conditions:

1. They are nonoptimal
 2. They are conserved in the eight analyzed *Aspergillus* species
 3. Their relative position in the protein is conserved in the eight analyzed *Aspergillus* species
- Codons 24 and 25 are the only ones which fulfill these three conditions. In the row of four adapted codons you mention, codons 26 and 27 are not conserved in all of the other *Aspergilli*, and then were not chosen. We believe that the changes introduced in figure 1 make our rationale more clear.

4) Line 125, “revised” by sharp et al. should be “reviewed” by Sharp et al.

Text was corrected

5) Line 197/198, should “non-repressing” better be “permissive”?

We prefer derepressing or non-repressing, since we are referring to transcriptional control. *ureA* is not inducible, but it is expressed in the absence of nitrogen catabolite repression (*i.e.* absence of ammonium). For details on the transcriptional regulation of *ureA*, please refer to Abreu *et al.* (2010).

6) Line 228/229, “mRNA levels are similarly diminished in the mutant strain” it should be clarified which of the mutant strains is referred to.

Text was corrected

Reviewer 2

Comments to the Author(s)

In this manuscript, Manuel Sanguinetti and co-authors attempted to demonstrate that a pair of nonoptimal codons, encoding Gln and Gly, at the junction of the N-terminal fragment and the first transmembrane helical segment of *Aspergillus nidulans* urea transporter, UreA, is important for the

biosynthesis of UreA and, presumably, its proper folding. The authors hypothesized that this pair of nonoptimal codons is essential for proper co-translational targeting of transmembrane segment(s) into ER membrane.

The authors tested their hypothesis by i) mutating these codons (individually or simultaneously) into synonymous optimal ones, ii) measuring UreA mRNA and protein levels, as well as urea uptake.

The authors found that mutation of just one of the two codons doesn't affect UreA biogenesis, while simultaneous mutation of the two codons affects UreA biogenesis. The observed effects were more pronounced at 37 deg C, but not at 24 deg C.

The authors argued that the pair of non-optimal codons "would presumably regulate translation rate in a very early stage in the process of biosynthesis, maybe contributing to the correct interaction of RNCs translating ureA with SRP and/or other factors necessary for these early events."

This study is interesting, although not novel and far too preliminary to be published in its present form.

First of all, we would like to point out a fundamental aspect of our work which we believe makes it especially interesting and novel. We cite the works by Fluman *et al.*⁽⁴⁾ and by Pechman *et al.*⁽⁵⁾, who show that mRNA-encoded pauses may be important for the interaction with SRP. However, there are significant differences between their results and ours.

Fluman *et al.* found that in *E. coli* pauses encoded in the mRNA were related to Shine-Dalgarno-like sequences. One of these pauses (codons 16 to 36) would occur before the emergence of the nascent polypeptide from the ribosomal tunnel. As observed in *E. coli*, according to our results codons 24 and 25 would exert their role while the nascent polypeptide has not yet emerged from the ribosome. But in the case of *ureA*, pausing would be determined by a pair of nonoptimal codons (instead of Shine-Dalgarno elements).

Note that what we observe is quite different from the results of Pechmann *et al.* who found that in *S. cerevisiae* nonoptimal codon clusters would establish pauses which would improve the recognition by SRP, but these would take place after the emergence of the SRP binding site in the nascent polypeptide.

Thus, in the case of *ureA*, a novel mechanism, not described before in eukaryotes might be operating. A couple of sentences were added to the manuscript in order to make this point more clear.

In *in vitro* studies in *S. cerevisiae* it has been demonstrated that SRP can recognize its substrates before its emergence from the ribosome⁽⁶⁾ and this early recognition would improve the efficiency of targeting and translocation to the ER membrane. After our results, we hypothesize that the elimination of the putative pause would cause an impairment in the early recognition of ribosomes translating *ureA* and hence a diminution in the cotranslational targeting to the ER membrane and its final destination to the plasma membrane. According to this hypothesis the overexpression of SRP would partially correct the observed mutant phenotype (see later).

Major:

(1) Apart from the "+70 pause hypothesis" originally suggested by François Képès 1996, it has been shown by Mullet and co-authors that ribosomes pause at specific sites during synthesis of membrane-bound chloroplast reaction center protein D1 and this pausing may facilitate co-translational folding of D1 and aid the integration of D1 into thylakoid membranes (see Kim et al, J Biol Chem. 1991; Kim et al, J Biol Chem. 1994, etc). The authors must cite these original observations.

We thank the reviewer for pointing out these interesting references, which we missed. We cited these papers in the Discussion of the revised version.

(2) The authors apparently presumed that synonymous substitutions under study would not lead to miscoding. While true in many cases, such possibility couldn't be excluded (see Drummond and

Wilke, Cell 2008; Buhr et al. Mol Cell, 2016, etc). The authors thus have to exclude miscoding by microsequencing of the silently mutated products.

We would like to point out the methodological handicaps to carry out the experiments the reviewer suggests, given the scarce amount of UreA2425 protein present at 37°C. We consulted Dr. Rosario Durán, head of the Analytical Biochemistry and Proteomics Unit in the Pasteur Institut of Montevideo, about the feasibility of performing microsequencing (or alternative techniques) on this low amount of protein and she strongly discouraged us. Moreover, the protein would have to be purified which, apart from the difficulties inherent to purification of membrane proteins, would be practically impossible considering the low synthesis levels of UreA2425. It must be taken into account that in order to avoid artifacts, the protein cannot be overexpressed and the culture conditions cannot be modified.

Anyway, for a number of reasons we do not think that the synonymous substitution of codons 24 and 25 could lead to miscoding. We are changing nonoptimal codons by optimal ones. According to the definition of an optimal codon, these are translated at higher speed and more accurately (Sharp et al.⁽⁶⁾) than their nonoptimal counterparts.

Finally, in the paper by Buhr *et al.* they harmonize the sequence of an eukaryotic gene for its expression in *E. coli*. Then, they introduce multiple synonymous substitutions, generating the HM variant, which has 38 synonymous substitutions throughout the coding sequence, some of which (we assume, since it is not clearly stated in the paper) substitute frequent codons by nonfrequent ones, and in other cases the reverse. We believe this is an extreme case, not comparable to our experimental system.

(3) Was the interaction with SRP indeed affected, as the authors hypothesized?

Some preliminary results of our group would support this hypothesis. Srp54 is the most conserved protein subunit of SRP, and the one in charge of direct recognition of the signal sequence in the peptide and of GTP-dependent recognition of the SRP-receptor in the ER membrane. We have overexpressed *srpA* (coding in *A. nidulans* for the ortholog of Srp54) by putting it under the control of the doxycycline-inducible tetON promoter. In these conditions the growth phenotype of *ureA2425* strains on urea and 2-thiourea disappears almost completely, and the strains behave as a wild-type (see figure R3). Since we intend to develop this research further, we have not wanted to include these results in the manuscript, but if you think it is necessary to incorporate them in the present work, we would agree to add a brief note on that.

Figure R3. The overexpression of *srpA* partially corrects the growth defect caused by the synonymous mutations of codons 24 and 25. Growth phenotypes of wild type and mutant UreA strains in the presence of doxycycline (1 µg/mL), on urea as nitrogen source or on 2-thiourea with sodium nitrate (NaNO₃) 10 mM as nitrogen source, at 37°C. In *wt-tetO srpA* and *ureA2425-tetO srpA* the expression of *srpA* is under the control of the Tet-On promoter. Growths on ammonium 5 mM and NaNO₃ 10 mM are used as controls. A *ureAΔ* strain is shown as negative control.

(4) The origin of the drastic reduction of the levels of the silently mutant UreA variant remains unclear and has to be determined.

We reason that the fact of having this drastic reduction in UreA levels can be due to lowered synthesis or augmented degradation. As we explain in the Discussion (and extended now after Reviewer's 1 concerns), none of our results points to misfolding and/or augmented degradation. We do not observe augmented vacuole degradation, accumulation or UreA-GFP in the ER, signs of UPR or proteasomal degradation. Hence, we believe that the drastic reduction of the levels of the mutant UreA must arise from impaired protein synthesis.

(5) The authors also have to appropriately cite recent reports from Dr. Elizabeth Grayhack's lab (see e.g. Ghoneim et al. Conservation of location of several specific inhibitory codon pairs in the *Saccharomyces sensu stricto* yeasts reveals translational selection. *Nucleic Acids Res.* 2019; Gamble et al. Adjacent Codons Act in Concert to Modulate Translation Efficiency in Yeast. *Cell.* 2016).

We thank the reviewer for signaling these interesting references. We have cited them in the discussion.

However, we would like to point out that the articles by Grayhack's group report on the identification in *Saccharomyces* species of a set of suboptimal codon pairs that substantially reduce the rate of translation elongation and the expression of the genes bearing them. The individual codons composing these pairs do not cause this effect. They showed that at least one of the codons in most of these inhibitory pairs is wobble decoded and that an interplay between tRNAs at adjacent sites in the ribosome would modulate translation efficiency. This phenomenon seems to be conserved in *Candida* species considered in the study. In the case of *ureA*, however, there are important differences with respects to Grayhack's results. To start with, they found that the codon pairs are inhibitory, while in the case of *ureA* the pair of nonoptimal codons 24 and 25 would be necessary for correct synthesis. Moreover, they found that in order to establish the inhibitory effect both codons in the pair are necessary, while in the case of *ureA* the presence of a single nonoptimal codon would be enough to grant normal synthesis.

(6) Some other recent reviews on the function of the rare codon clusters should be mentioned as well (the Chaney JL, Clark PL. Roles for Synonymous Codon Usage in Protein Biogenesis. *Annu Rev Biophys.* 2015; Komar AA. The Yin and Yang of codon usage. *Hum Mol Genet.* 2016).

Cites have been introduced

Minor:

1. The authors should attempt to write the paper in a more concise and clear way and carefully proofread it.

The manuscript has been revised and rephrased

2. While describing the location of the rare codon cluster under study (within the UreA mRNA), the

authors should use the term “boundary” (as they did on page 13; line 279), or border, or junction, rather than “limit”.

Text was corrected

3. Page 7, line 125: “(revised by Sharp et al. [37]).” I am assuming the authors meant to write “reviewed by”.

Text was corrected

References

- (1) Gournas, C. et al. Transport-dependent endocytosis and turnover of a uric acid-xanthine permease. *Mol Microbiol* 75:246-260 (2010).
- (2) Abreu, C. et al. UreA, the major urea/H⁺ symporter in *Aspergillus nidulans*. *Fungal Genet. Biol.* 47, 1023–1033 (2010).
- (3) Evangelinos M. et al. BsdA Bsd2 -dependent vacuolar turnover of a misfolded version of the UapA transporter along the secretory pathway: prominent role of selective autophagy. *Mol. Microbiol.* 100, 893–911 (2016).
- (4) Fluman N. et al. mRNA-programmed translation pauses in the targeting of *E. coli* membrane proteins. *Elife* 3. (2014).
- (5) Pechmann S. et al. Local slowdown of translation by nonoptimal codons promotes nascent-chain recognition by SRP in vivo. *Nat. Struct. Mol. Biol.* 21, 1100–1105 (2014)
- (6) Berndt U. et al. A signal-anchor sequence stimulates signal recognition particle binding to ribosomes from inside the exit tunnel. *Proc. Natl. Acad. Sci.* 106, 1398–1403 (2009).